# Influence of atmospheric internal variability on the long-term Siberian water cycle during the past two centuries

Kazuhiro Oshima[1], Koto Ogata[2,3], Hotaek Park[1], Yoshihiro Tachibana[2]

[1] Institute of Arctic Climate and Environment Research, Japan Agency for Marine-Earth Science and Technology, Yokosuka, Japan
[2] Weather and Climate Dynamics Division, Mie University, Tsu, Japan
[3] Aerological Observatory, Japan Meteorological Agency, Tsukuba, Japan

*Correspondence to*: Kazuhiro Oshima (kazuhiroo@jamstec.go.jp)

**Abstract.** River discharges from Siberia are a large source of freshwater into the Arctic Ocean, whereat the cause of the long-term variation in Siberian discharges is still unclear. The observed river discharges of the Lena in the east and the Ob in the west indicated different relationships in
each of the epochs during the past seven decades. The correlations between the two river discharges were negative during the 1980s to mid-1990s, positive during the mid-1950s to 1960s, and became weak after the mid-1990s. More long-term records of tree-ring-reconstructed discharges have also shown differences in the correlations in each of the epochs. It is noteworthy that the correlations obtained from the reconstructions tend to be negative during the past two centuries. Such tendency has also been obtained from precipitations in observations, and in simulations with an atmospheric general circulation model (AGCM) and fully coupled atmosphere-ocean GCMs conducted for the Fourth Assessment Report of the
IPCC. The AGCM control simulation further demonstrated that an east–west seesaw pattern of summertime large-scale atmospheric circulation frequently emerges over Siberia as an atmospheric internal variability. This results in an opposite anomaly of precipitation over the Lena and Ob and the negative correlation. Consequently, the summertime atmospheric internal variability of east–west seesaw pattern over Siberia is a key factor influencing the long-term variation in precipitation and river discharge, i.e., the water cycle in this region.

## 1 Introduction

The river discharge ($R$) from the pan-Arctic terrestrial area supplies freshwater, nutrients, and organic matter to the Arctic Ocean. The three great Siberian rivers, the Lena, Yenisei and Ob (Figure 1) account for about 60% of the total $R$ into the Arctic Ocean and have an important role in the freshwater budget and climate system in the Arctic (e.g., Aagaard and Carmack, 1989, 1994). Numerous studies have investigated the interannual variation and linear trend of the Siberian $R$ (e.g., Berezovskaya, et al., 2004; Ye et al., 2004; McClelland et al., 2004, 2006; Rawlins et al., 2006; MacDonald et al., 2007; Shiklomanov and Lammers, 2009), however they have mainly analyzed the $R$ dataset from a hydrological

perspective. Several other studies have been conducted to determine the linkages among atmospheric circulation, moisture transport, precipitation (*P*), precipitation minus evapotranspiration (*P-E*), and the *R* for Siberian rivers using atmospheric reanalysis combined with the *R* dataset (Fukutomi et al., 2003; Serreze et al., 2003; Zhang et al., 2012; Oshima et al., 2015). To understand such linkages, it is necessary to improve our knowledge of the atmospheric and terrestrial water cycles in the region.

5  Theoretically, *P-E* over a basin, which is the net input of water from the atmosphere to the land surface, corresponds to *R* at the river mouth as a long-term average. Indeed, they quantitatively agree well for the individual Siberian rivers (e.g., Zhang et al., 2012; Oshima et al., 2015). The *R* and *P-E* are strongly affected by the *P* and associated atmospheric moisture transport over the individual regions. Processes of the atmospheric moisture transport associated with the *P-E* show regional difference among the Siberian rivers (Oshima et al., 2015). The *P-E* over the Lena is mainly supplied by a transient moisture flux associated with cyclone activity and that over the Ob is mainly supplied by a stationary moisture flux

10 associated with seasonal mean wind. Both processes affect the area over the Yenisei.

  Regarding the interannual variations, the moisture transport, *P-E*, *P*, and *R* also relate to each other, while those relationships have some seasonal time lag due to the large area of the basin, snow accumulation in winter, negative or near zero *P-E* in summer and terrestrial processes (e.g., discharge control via dams, permafrost condition associated with runoff process, distributions of lake, wetland and vegetation associated with evapotranspiration) as discussed in Oshima et al. (2015). More details about this are given in the last part of next section. Fukutomi et al.

15 (2003) elucidated that the interannual variation in summer *P* over the Lena was negatively correlated with that over the Ob during the 1980s to mid-1990s. The summer (*P-E*)s of the two rivers and corresponding autumn *R*s , respectively, were also negatively correlated in the same period. Furthermore, Fukutomi et al. (2003) indicated that the negative correlations were affected by an east–west seesaw pattern of large-scale atmospheric circulation and associated moisture transport over Siberia. When the cyclonic anomaly of atmospheric circulation emerges over the Lena river, the simultaneous anticyclonic anomaly emerges over the Ob river. The cyclonic anomaly induces a convergence of moisture flux over

20 the Lena basin, then increases *P* and *R* of the Lena river. In contrast, the anticyclonic anomaly over the Ob induces a divergence of moisture flux, then decreases *P* and *R* of the Ob river, and vice versa. Thus, the east–west seesaw pattern produced the negative correlation of *R*s/*P*s between the Lena and Ob during the 1980s to mid-1990s. While the influence of cyclone activity on the interannual variations in *P-E* and *R* was discussed in their studies (Fukutomi et al., 2004, 2007, 2012), the cause of the negative correlations has not been fully explained, and it is not certain whether the negative correlation occurs in other periods.

25  The negative correlation noted above was apparent during the 1980s to mid-1990s. More recently, several drastic changes in the terrestrial water cycle have occurred around Yakutsk in eastern Siberia. Increases in *P* and soil moisture, and deepening of the active layer (Ohta et al., 2008, 2014; Iijima et al., 2010; Iwasaki et al., 2010) have been observed, particularly during 2005–2008, and the wet conditions have induced flooding (Fujiwara, 2011; Sakai et al., 2015) and forest degradation (Iwasaki et al., 2010; Iijima et al., 2014; Ohta et al., 2014). Moreover, effects of

permafrost degradation on changing thermokarst lakes and landscapes have been reported in the last two decades (Fedorov et al., 2014). While these are local changes, the observed results suggest that some changes on a large spatial scale also occurred in this region in recent decades. Indeed, Iijima et al. (2016) showed that the increase in $P$ and the wet conditions in eastern Siberia during the mid-2000s were affected by cyclone activity accompanied by changes in large-scale atmospheric circulation over Siberia. This suggests that the relationship between the Lena and Ob, which was negative correlation during the 1980s to mid-1990s, recently changed. However, the long-term variation and its effects on the water cycle in this region are still unclear.

To examine the long-term variation in $R$ of the Lena and Ob Rivers, in addition to the observed $R$ during the past seven decades, we analyzed reconstructed $R$ based on tree rings during the past two centuries. We investigated whether the negative correlation of $R$ between the Lena and Ob occurred before 1980s. We further examined an influencing factor on the long-term variation in $R$ and $P$, and the associated atmospheric circulation using atmospheric reanalyses and simulations with an atmospheric general circulation model (AGCM) and atmosphere-ocean coupled models archived in the World Climate Research Programme's Coupled Model Intercomparison Project phase 3 (CMIP3, Meehl et al., 2007).

## 2 Data and analysis methods

Monthly $R$ observed near the river mouths of the Lena and Ob (i.e., Kusur and Salehard, Figure 1) from the Arctic-Rapid Integrated Monitoring System for the period of 1936–2009 (http://rims.unh.edu/), and annual $R$ reconstructed based on tree rings for the period of 1800–1990 (MacDonald et al., 2007, http://onlinelibrary.wiley.com/doi/10.1029/2006JG000333) were used. While the negative correlation was seen during the 1980s to mid-1990s, the time scale of the negative correlation seems one or two decades. To detect a robust tendency of the correlation, we made subsets of the dataset and increased sample size of data. In addition to the entire period, we analyzed subsets of 150-year periods for the reconstructed $R$. There is a 191-year record of reconstructed $R$, and we produced 5 subsets of 150-year records, with the start years delayed successively by one decade.

Monthly $P$ from the Global Precipitation Climatology Center (GPCC, Schneider et al., 2013) was compared to the $R$. While we used here the GPCC product, it has been confirmed that the $P$ from the other products (e.g., PREC/L: Chen et al. 2002, APHRODITE: Takashima et al. 2009; Yatagai et al. 2012) also have strong positive correlation with $R$ for the Lena and Ob Rivers (Oshima et al. 2015). For simplicity, we defined the area of 50–70°N and 110–135°E as the Lena region, and the area of 50–70°N and 60–85°E as the Ob region. The area averaged $P$ over these regions corresponded well with the averages over the individual river basins. The correlations during 1901-2010 were 0.89 for the Lena and 0.86 for the Ob. In analyses of atmospheric circulation, geopotential height at 500 hPa ($Z500$) from two atmospheric reanalyses, the Japanese 55-year Reanalysis (JRA-55, Kobayashi et al., 2015; Harada et al., 2016) and the National Oceanic and Atmospheric Administration-Cooperative Institute

for Research in Environmental Sciences (NOAA/CIRES) Twentieth Century Reanalysis (20CR, Compo et al., 2011), was used. The time period of the $P$ and $Z500$ datasets was from 1901 to 2010, except for the JRA-55, which started from 1958.

There are long-term records of tree-ring-reconstructed $R$s over the past two centuries, whereas the meteorological data are limited to the 20th century. To examine the long-term variations and intrinsic atmospheric circulation (i.e., internal variability, teleconnection and feedback) associated with the $P$, a 300-year control simulation was performed with an AGCM developed by the Center for Climate System Research, University of Tokyo, and the National Institute for Environmental Studies (Numaguti et al., 1995, 1997). The setting of the control simulation is the same as in Ogata et al. (2013). The horizontal resolution is about 300 km and the vertical discretization comprises 20 layers (T42L20). It started from a state of rest with constant temperature, and was forced by the climatological seasonal cycle of sea surface temperature (SST), sea ice, and fixed greenhouse gases (GHG) as boundary conditions. We excluded the first 5 years of data from the 300-year simulation as the spin-up time. For the AGCM control simulation, we made 15 subsets of 150-year records, with the start years delayed successively by one decade. As in Numaguti (1999) and Kurita et al. (2005) based on the same AGCM with the same horizontal resolution, the spatial pattern and seasonal cycle of simulated $P$ and atmospheric circulation over Siberia are generally consistent with the observed features in the seasonal timescale.

In addition, control simulations under pre-industrial conditions (PICTL) and "the 20th century climate in coupled models" (20C3M) simulations in the CMIP3 multi-models conducted for the Fourth Assessment Report of the Intergovernmental Panel on Climate Change (IPCC AR4, Meehl et al., 2007, IPCC 2007) were compared to the AGCM control simulation. The 20C3M and PICTL simulations were forced by the GHG increasing as observed through the 20th century and the constant pre-industrial levels of GHG, respectively. While the time periods of the CMIP3 simulations were different among the models, the 20C3M simulations were from 1850–1900 to 2000–2001. The PICTL simulations had time records from 81 to 1001 years. We analyzed the PICTL simulations that were longer than 150 years and made subsets of 150-year records with the start years delayed successively by five decades in each of the PICTL simulations. All of the 23 models with the multi-ensemble members in the CMIP3 simulations under the PICTL and 20C3M scenarios were used.

While the reconstructed $R$ comprises an annual value, we analyzed seasonal mean values for the observed $R$, $P$, and $Z500$. Because there is a seasonal time lag between $P$ and $R$ for the Lena and Ob Rivers, and also the atmospheric circulation and $P$ in these regions have large seasonality. As in Tachibana et al. (2008) for the Amur River and Arpe et al. (2014) for the Volga River, it is expected that the summer $P$-$E$ may correspond to autumn $R$, and the summer $P$-$E$ and $P$ are governed by atmospheric circulation in summer. Using a similar method of Tachibana et al. (2008) ad Oshima et al. (2015), we compared all possible combinations of seasonal averaging period for $P$-$E$ and $R$. As a result, a pair of summer period from June to September and autumn period from August to October showed high correlation between summer $P$ and autumn $R$, and is best match for the Lena and Ob Rivers. The correlations during 1936–2009 are 0.79 for the Lena and 0.64 for the Ob, both significant above the 99% confidence level (Table 1). In addition, due to the large amount and large variability of water vapor in summer, it is expected that the interannual

variations in summer $P$ and corresponding autumn $R$ dominate the annual values. While those were still indicated in the previous studies (Fukutomi et al., 2003; Zhang et al., 2012), we confirmed the contribution of seasonal values of $P$ and $R$ to annual values. The correlation between the summer $P$-$E$ (autumn $R$) and its annual value is 0.91 (0.79) for the Lena, and that for the Ob is 0.64 (0.91). Therefore, we used the summer $P$ and $Z500$ averaged from June to September and autumn $R$ averaged from August to October in the analysis.

## 3 Results

### 3.1 Long-term variation

### 3.1.1 Observed and reconstructed river discharges

Figure 2a shows the time-series of observed autumn $R$ at the river mouths of the Lena (red solid line) and Ob (red dashed line) during the past seven decades (1936–2009), with 15-year running correlations between them (black line). Although the correlations were strong and negative during the 1980s to mid-1990s as in Fukutomi et al. (2003), those were positive during the 1950s to 1960s and became weak after the 1990s. As mentioned above, these autumn $R$s correspond to the summer $P$s. The time-series of the summer $P$ over the Lena and Ob regions (Figure 2b) indicate a negative correlation around the 1910s, during the 1940s to mid-1950s, and after the 1980s. The correlations of $P$ were near zero in the 1920s, and were weak and positive during the 1960s. While there were some differences between the observed $R$ and $P$, the $P$ displayed a strong negative correlation in the 1980s and positive correlation in the 1960s. These results from the observations indicate that the relationship of $R$/$P$ between the Lena and Ob was different in each of the epochs.

Figure 2c shows a long-term time-series of tree-ring-reconstructed annual $R$ of the Lena and Ob during the past two centuries (1800–1990). Similar to the observations, the correlations of reconstructed $R$ were negative during the 1980s to mid-1990s and positive during the 1950s to 1960s, while there was some discrepancy between the observed $P$ and reconstructed $R$ in the early 20th century. The discrepancy may be due to error and uncertainty both in the observation and reconstruction. The observation stations of $P$ are sparse in Siberia and measuring $P$ is difficult such as wind-induced undercatch, wetting, and evaporation losses. While the reconstructed $R$ is based on the tree-ring width, the tree-ring width has an indirect relationship with the $R$ and the both are mainly related through the $P$. There are also other influences such as air temperature, solar radiation, and nitrogen. In addition, the tree-ring width is affected by meteorological conditions during the growing season in summer and there must be less contribution from the conditions during winter. As a result, the reconstructed $R$s can explain 43% of the observed variability for the Lena and 51% for the Ob (MacDonald et al. 2007). In the 19th century, the correlations of reconstructed $R$ were strong and negative in some epochs (1810s, 1850s, and 1890s) and moderate or weak and positive in some other epochs (1880s and 1900s). These results also indicate that the relationship between the Lena and Ob differed in each of the epochs. However, it is noteworthy that negative correlations were frequently seen in

the time-series of reconstructed $R$ (black line in Figure 2c). As shown by the red bar histogram in Figure 3a, many of the correlations for reconstructed $R$ were negative. The correlations of observed $R$ and $P$ also tended to have negative values, although these results may not be as robust due to relatively short records (observed $R$: 74 years, $P$: 111 years). It is considered that the long-term change on decadal timescale or long-term trend may affect the correlations in Figure 2. While Fukutomi et al. (2003) and MacDonald et al. (2007) discussed about the long-term variations on decadal timescale, it seems that the long-term changes do not affect the time series of the correlations. Indeed, when we remove the 19-year running mean from the raw time-series of $P$ and $R$ in Figure 2a-c, the correlations do not change so much (not shown) and there is the tendency of frequent negative correlation. To quantitatively show a tendency of the correlation, we calculated median and skwness as a metric of the frequent distribution of the correlations. The skewness is a measure of asymmetry of frequency distributions. When the frequent distribution is distributed in the negative (positive) side, the skewness has positive (negative) value. As a result, the medians of the 15-year running correlations in the observeion and reconstruction were negative and their skewnesses were positive, although the skewness of observed $P$ was nearly zero (Figure 3b and Table 2). Therefore, the interannual variation in $R$s/$P$s of the Lena and Ob Rivers has tended to be out-of-phase during the past two centuries. This may suggest that the east–west seesaw pattern frequently emerges over Siberia.

### 3.1.2 Simulated precipitation

To determine the intrinsic atmospheric circulation associated with the variation in summer $P$, we analyzed the AGCM control simulation. As with the reconstructed $R$, the correlations of simulated summer $P$ between the Lena and Ob regions were largely negative. The histogram of the correlations of simulated $P$ was distributed in the negative side (blue line in Figure 3a), the median was negative and the skewness was positive (blue cross markes in Figure 3b and Table 2). Compared to the reconstructed $R$, the distribution of simulated $P$ was more negative than positive (Figure 3a) and the median and skewness from the simulated summer $P$ (Table 2) tended to be more negative and positive, respectively (Figure 3b). The results indicate that atmospheric internal variability in summer leads to the negative correlation of summer $P$. The AGCM control simulation has no external forcing, and boundary conditions such as SST, sea ice, solar activity, and GHG are fixed. Consequently, the variation in simulated $P$ and $Z500$ in the control simulation can be interpreted as internal variability in the model.

The 20C3M and PICTL simulations in the CMIP3 coupled models provided more evidence for intrinsic atmospheric variability, including air–sea interactions. The medians and skewness of the correlations of summer $P$ between the Lena and Ob regions in the CMIP3 simulations were plotted in Figure 3b (black and gray cross marks); the plotted marks were largely distributed in the upper-left side and the median and skewness also tended to be negative and positive, respectively, while they were well scattered. This suggests that some models failed to reproduce the summer $P$ variability and atmospheric circulation over Siberia. However, note that many simulation results were plotted around the tree-ring-reconstructed $R$ and most results from the CMIP3 simulations were distributed toward the center compared to those from the AGCM control

simulation (Figure 3b). These results imply some effects of air–sea interactions on the $P$ variability over the Lena and Ob. This is discussed in the final section.

As a result, similar to the observation and reconctuction, the AGCM and CMIP3 simulations demonstrated that the $P$ over the Lena and Ob tends to be out-of-phase. While there were weak and positive correlations of summer $P$ in several periods (Figures 2 and 3a), we focused on the

negative correlation and further examined summertime atmospheric circulation pattern associated with the $P$ over Siberia.

### 3.2 Atmospheric circulation associated with the negative correlation of precipitations

To identify summertime dominant atmospheric circulation patterns associated with summer $P$ variability, we performed an empirical orthogonal function (EOF) analysis on summer $Z500$ over the three great Siberian river basins (blue inset box in Figure 4). The spatial pattern of

the first EOF mode (EOF1) is the cyclonic circulation anomaly centered in the vicinity of the coast in central Siberia (not shown). This pattern only enhances the eastward moisture transport over Siberia, and the effect on moisture convergence/divergence over the Lena and Ob regions is small. The EOF2 indicated an east–west seesaw pattern similar to Fukutomi et al. (2003). While Figure 4 is the spatial pattern of EOF2 based on the JRA-55, the result of 20CR showed a similar pattern, for which the pattern correlation was 0.89. This seesaw pattern of EOF2 directly affects moisture convergence and divergence over the two river basins and results in changes in the $P$ over the regions.

To confirm the effects of the east–west seesaw pattern on the $P$, we compared the difference in $Z500$ over the western and eastern Siberia regions (west–east difference in $Z500$: $\Delta Z500_{WE}$) and the difference in $P$ over the Lena and Ob regions (Lena–Ob difference in $P$: $\Delta P_{LO}$). We defined the Lena and Ob regions for $P$ (green inset boxes in Figure 4), which cover almost all of the basins, while the regions for the $Z500$ were shifted 10° westward (purple inset boxes), which covered almost all of the negative and positive centers of action of EOF2. As described in the Introduction, when $Z500$ anomalies are negative over the east and positive over the west as shown in Figure 4, $P$ anomalies must be positive over

the Lena region and negative over the Ob region. As expected, $\Delta P_{LO}$ was positively correlated with $\Delta Z500_{WE}$. The correlation coefficients were 0.72 for the JRA-55 and 0.60 for the 20CR, both significant above the 99% confidence level (Figure 5).

Similar results (i.e., the east–west seesaw pattern of EOF2 and the positive correlation between the $\Delta P_{LO}$ and $\Delta Z500_{WE}$) were obtained in the AGCM control simulation and in the 20C3M and PICTL simulations from the CMIP3 coupled models, while some CMIP3 simulations failed to reproduce these features. The pattern correlation of the EOF2 patterns between the JRA-55 and AGCM was 0.83. The pattern correlations with

JRA-55 for the 20C3M and PICTL simulations ranged from -0.62 to 0.94, but those in 81% of the 20C3M and 76% of the PICTL simulations were greater than 0.7. Several CMIP3 models simulated the seesaw pattern in the EOF3. These results from the AGCM and CMIP3 simulations indicated that the seesaw pattern emerges as a dominant mode of the summertime atmospheric circulation over Siberia. The correlation between

the $\Delta P_{LO}$ and $\Delta Z500_{WE}$ in the AGCM was 0.55 for the entire period of the 295-year record and 0.53–0.63 for the 15 subsets of 150-year records. The correlations between the $\Delta P_{LO}$ and $\Delta Z500_{WE}$ in 94% of the 20C3M and 90% of the PICTL simulations were greater than 0.7. The above results in the simulations also indicated that the east-west seesaw pattern is related with the negative correlation of $P$.

Therefore, the results of the simulations with the AGCM and CMIP3 models were basically consistent with the reconstructed $R$ and
observations, and they support the linkage between the summertime east–west seesaw pattern over Siberia and the out-of-phase $P$ over the Lena and Ob regions.

## 4 Summary and discussion

We examined the long-term variations in the $R$s and corresponding $P$s for the Lena in eastern Siberia and the Ob in western Siberia based on observations, tree-ring reconstructions, and simulations with the AGCM and CMIP3 models. The observations during the past seven decades
indicated that correlations of observed $R$s between the Lena and Ob were negative during the 1980s to mid-1990s as in Fukutomi et al. (2003), but positive during the mid-1950s to 60s and became weak in recent decades (Figure 2a). This suggests that the relationship between the Lena and Ob $R$s was different in each of the epochs. However, the reconstructed $R$s during the past two centuries indicated that the Lena and Ob tended to be negatively correlated, i.e., out-of-phase (Figures 2c and 3). The observed $P$s over eastern and western Siberia also frequently had negative correlations in the 20th century (Figure 2b), which were affected by the east–west seesaw pattern of summertime atmospheric circulation over
Siberia (Figure 4). Compared to the reconstructed $R$ and observed $P$, the simulated $P$ in the AGCM control simulation indicated more frequent negative correlations in association with the seesaw pattern (Figure 3). Because of the fixed boundary conditions, the control simulation demonstrated that the negative correlation and the seesaw pattern emerge as summertime atmospheric internal variability over Siberia. Although the results from the 20C3M and PICTL simulations vary among the models, they basically support the above features. As a consequence, the east–west seesaw pattern of large-scale circulation frequently emerges as summertime atmospheric internal variability over Siberia and induces
the convergence/divergence of moisture flux and associated opposite anomaly, i.e. negative correlation, of the summer $P$s over eastern and western Siberia, resulting in the out-of-phase autumn $R$s of the Lena and Ob Rivers. Therefore, the summertime atmospheric internal variability of the seesaw pattern over Siberia is a key factor influencing the water cycles in this region.

The results from the AGCM and CMIP3 simulations and previous studies give us further implication for the $P$ variability and associated atmospheric circulation pattern over Siberia. Compared to the AGCM control simulation, the CMIP3 simulations mostly plotted around the
reconstructed $R$ (Figure 3b), suggesting that the air–sea interaction acts as a damping of the seesaw pattern and breaks the negative correlation of $P$. An external forcing such as a SST or sea ice anomaly may affect large-scale circulation and $P$ over Siberia. Moreover, while the negative correlation dominated in the $P$ variations between eastern and western Siberia, the positive and weak correlation periods were also seen in some

periods as shown by the time-series in Figure 2. This implies that, in addition to the east–west seesaw pattern of atmospheric internal variability, there are other effects on the summertime $P$ variability over Siberia. Indeed, Sun et al. (2015) reported the remote influence of Atlantic multidecadal variation, which is an oscillation of North Atlantic SST between basin-wide uniform warm and cold conditions, on the variation in summertime $P$ over Siberia on decadal or multidecadal timescales. Iwao and Takahashi (2006, 2008) indicated that the effects of quasi-stationary

5   Rossby waves originated from blocking anticyclones in the North Atlantic–European sector on the precipitation seesaw pattern between northeast Asia and eastern Siberia. Ding and Wang (2005) showed a circumglobal teleconnection with zonal wavenumber-5 structure in the Northern hemisphere mid-latitude, resulting in $P$ anomalies in various areas of the world including Siberia. Iijima et al. (2016) indicated the impact of enhanced storm activity on the increase in $P$ and permafrost degradation in eastern Siberia during the mid-2000s and they discussed the relationship with the Arctic dipole anomaly associated with the sea ice reduction. As in Iijima et al. (2016), Fujinami et al. (2016) and Hiyama et

10  al. (2016) also showed the similar result for the $P$ over eastern Siberia. While they studied somewhat different timescales and different regions, the $P$ variability over the Lena and Ob must be affected by a combination of these processes including internal variability. However, this study did not examine those specific effects and future work is needed. In addition, it seems that the differences between the 20C3M and PICTL simulations are not large (Figure 3b), and there should be no significant influence of changes in GHG on the $P$ variability in Siberia, while $P$ in future projections will increase under global warming (IPCC, 2007, 2013).

**Acknowledgments**

This work was supported partly by the JSPS KAKENHI Grant Number 24241009 and 26340018, the GRENE Arctic Climate Change Research Project, the Arctic Challenge for Sustainability (ArCS) Project and the Joint Research Program of the Japan Arctic Research Network Center.

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

**Table 1:** Correlation coefficients among the summer $P$, annual $P$, autumn $R$, and annual $R$ for (a) the Lena and (b) Ob Rivers during 1936–2009. Summer (autumn) averaging period is from June to September (from August to October). The $P$ and $R$ are based on the Arctic-RIMS and GPCC. All values are above the 99% confidence level. Bold values are specifically described in the text.

| **Lena** | Summer $P$ | Annual $P$ | Autumn $R$ | Annual $R$ |
|---|---|---|---|---|
| Summer $P$ | 1.00 | **0.91** | **0.79** | 0.66 |
| Annual $P$ | | 1.00 | 0.72 | 0.73 |
| Autumn $R$ | | | 1.00 | **0.79** |
| Annual $R$ | | | | 1.00 |
| **Ob** | Summer $P$ | Annual $P$ | Autumn $R$ | Annual $R$ |
| Summer $P$ | 1.00 | **0.64** | **0.63** | 0.57 |
| Annual $P$ | | 1.00 | 0.47 | 0.57 |
| Autumn $R$ | | | 1.00 | **0.91** |
| Annual $R$ | | | | 1.00 |

5   **Table 2:** Median and skewness of the 15-year running correlations for the tree-ring-reconstructed annual $R$ (Figure 2c), observed autumn $R$ (Figure 2a), observed summer $P$ (Figure 2b), and simulated summer $P$. The observed $P$ and simulated $P$ are based on the GPCC and AGCM. A histogram and scatter diagram for these values are shown in Figure 3a and 3b. Values in brackets indicate the range of statistics calculated from 5 (15) subsets of 150-year records for the reconstructed $R$ (simulated $P$).

|  | Median | Skewness |
|---|---|---|
| **$R$_tree-ring** | **-0.25**<br>(-0.24 to -0.19) | **0.52**<br>(0.23 to 0.44) |
| $R$_obs. | -0.24 | 0.33 |
| $P$_GPCC | -0.32 | -0.02 |
| **$P$_AGCM** | **-0.36**<br>(-0.44 to -0.28) | **0.79**<br>(0.55 to 1.06) |

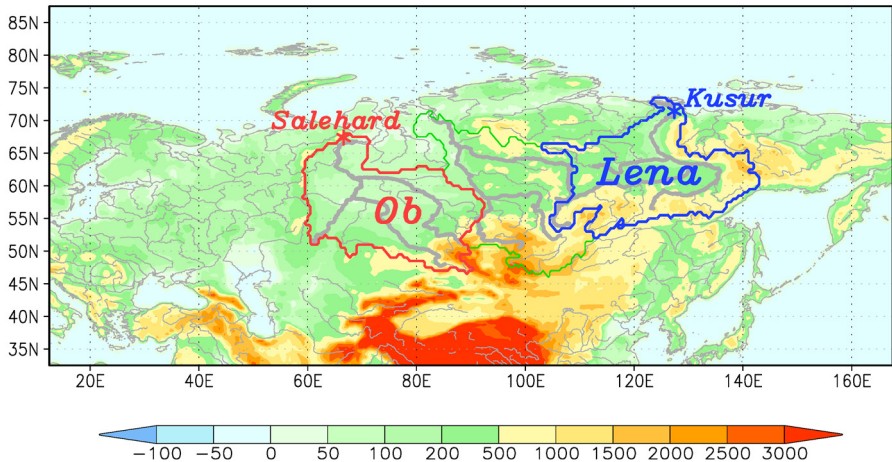

**Figure 1:** Map of study area of Siberia. The colored solid contours show the boundaries of each river basin (Lena: blue and Ob: red). The asterisks denote the locations of Kusur and Salehard, which are the observation stations nearest the river mouths. The color shades and thick gray lines denote elevation and major flow paths, respectively.

**Figure 2:** Time series of (a) observed autumn *R* during 1936–2009, (b) observed summer *P* during 1901–2010, and (c) tree-ring-reconstructed annual *R* during 1800–1990 of the Lena and Ob Rivers. Red (Blue) solid and dashed lines denote the *R*s (*P*s) of the Lena and Ob, respectively. Black thick lines denote 15-year running correlations between the Lena and Ob *R*s/*P*s. The confidence levels at 90%, 95% and 98% for the 15-year correlation are 0.44, 0.51 (yellow lines) and 0.59. Note that the axes of the *R* and *P* are shown on the left side of the panel and the axes of the correlations are shown on the right side.

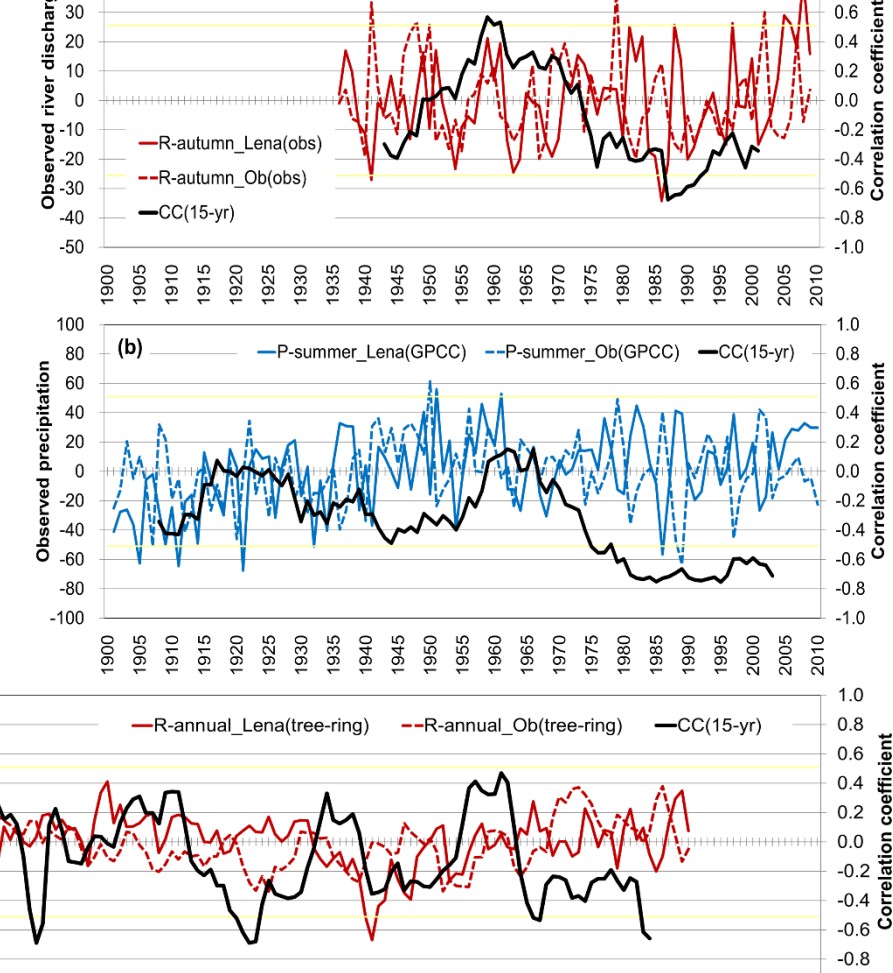

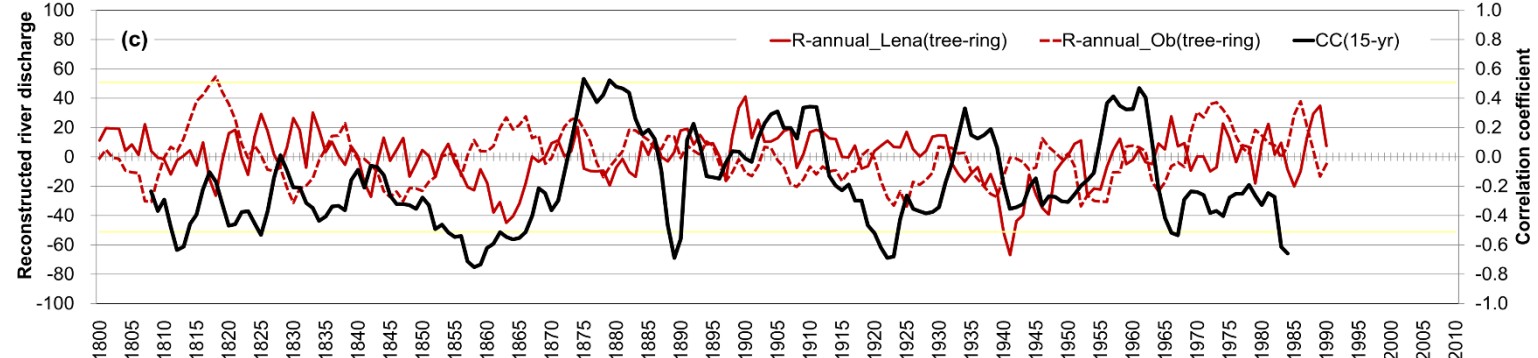

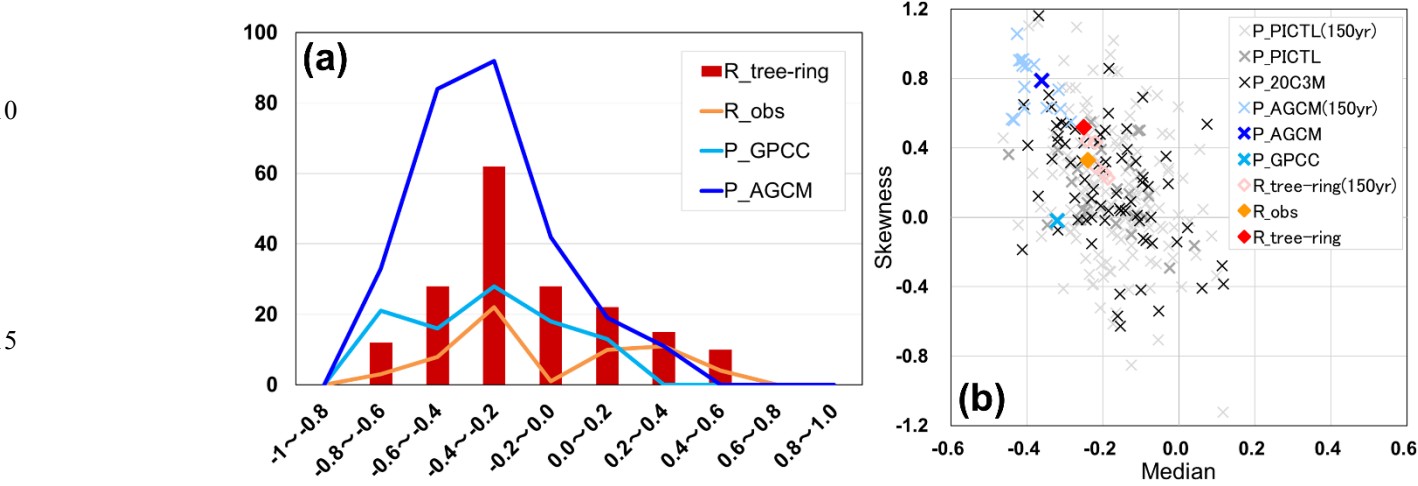

**Figure 3:** (a) Histogram of 15-year running correlations from the tree-ring-reconstructed annual *R* (red bars), observed autumn *R* (orange line), observed summer *P* (light blue line), and AGCM simulated summer *P* (blue line). (b) Scatter diagram between median and skewness of each of the 15-year correlations. Simulated *P* in the CMIP3 models' simulations (20C3M and PICTL), and subsets of 150-year record for the reconstructed *R* (5 samples), AGCM simulated *P* (15 samples), and PICTL simulated *P* (over 100 samples) are also plotted.

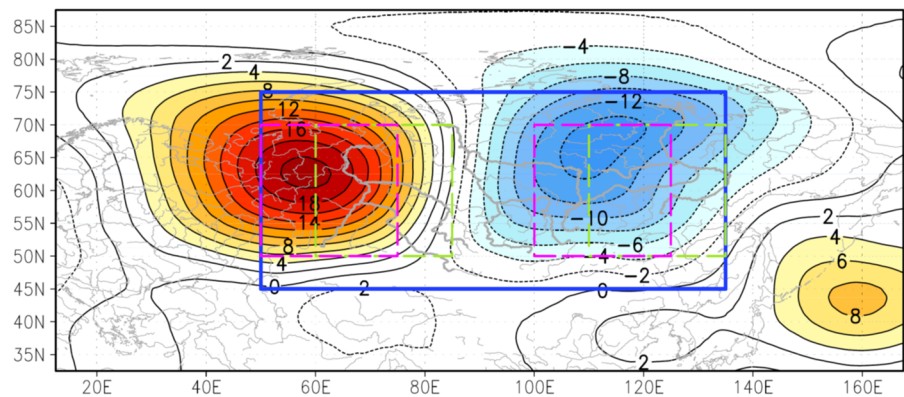

**Figure 4:** Spatial pattern of EOF2 (19.9% of explained variance) for the summertime *Z500* over Siberia region (blue line inset box: 45–75°N, 50–135°E, covering the three great Siberian rivers). Green (magenta) dashed inset boxes cover almost all areas of the Lena and Ob river basins (western and eastern centers of action of EOF2). The EOF analysis is based on JRA-55.

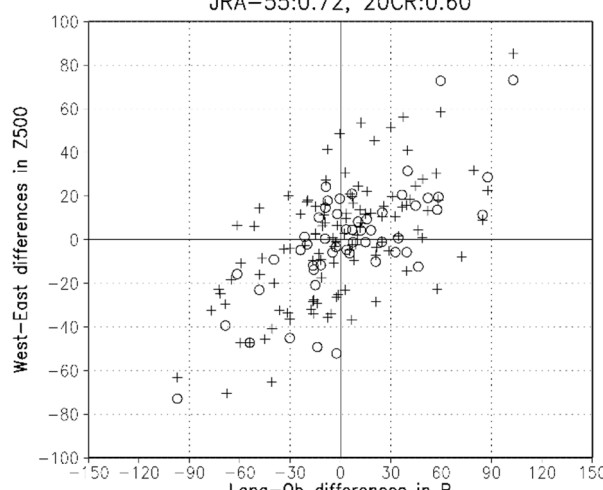

**Figure 5:** Scatter plot of the summer $P$ differences between the Lena and Ob regions ($\Delta P_{LO}$) and the summer $Z500$ differences between the western and eastern Siberia regions ($\Delta Z500_{WE}$). The areas of $\Delta P_{LO}$ ($\Delta Z500_{WE}$) are defined as green (purple) dashed inset boxes in Figure 4. The $\Delta Z500_{WE}$ based on JRA-55 and 20CR are plotted as marked with circles and crosses. Correlation coefficients between $\Delta P_{LO}$ and $\Delta Z500_{WE}$ are shown in the upper side of the scatter plot.