# Peer review of "Influence of atmospheric internal variability on the long-term Siberian water cycle during the past two centuries"

_Earth System Dynamics, 2017_

## Referee Comment (RC1) · K. Arpe (Referee) · 19 Oct 2017

The paper tries to find connetions between the river discharges of 2 main rivers in Siberia and the atmospheric circulation. It is well written and I enjoyed reading it and found is inspiring. I would recomment accepting it for printing it even if it far from reflecting a breakthrough in science.

I would add a plot in the introduction of topograpy and catchment areas to set the scene, like the attached file Fig_K1_oro. The Fig.1 provided in the manuscript, showing nearly the same, has too much information and the essentials are not easily to be seen.

[Figure]

I further suggest to add some Teleconnection maps (correlation maps) between the discharge or the precipitation over the catchment areas and the precipitation at each grid point as attached File figK2.jpg and FigK3.jpg for the Lena and figK4.jpg and figK5.jpg for the Ob. as suggested in the manuscript, I averaged the precipitation from June to September for such a comparison with the August to November river discharge. It shows nicely the catchment areas (figK2 and FigK4), though the higher correlations between river discharge and precipitation extend a little further, for the Lena to the SE and for the Ob beyond the Ural mountains but no connections between the 2 rivers. I am using the anomaly correlation, i.e. taking away a long term mean from each time series because in meteorology we are mostly interested in the deviations from the climatological mean. For convenience I use the mean of the whole time series provided. The values are given in the plots in % for clarity with less digits to be printed.

I wonder if the definition of correlation has changed during the last 40 years, since I wrote most of my programs. Modern papers show always very high values of > 0.9 while here the best values are> 0.7 and in most of my publications I am happy to reach correlations of up to 0.4.

I tried also to reproduce the anti-correlation between the Ob and Lena in the 1970/1980s, but could not find anythng in the precipitation like that although Fig.2b of the manuscript shows quite a few events like that. Plotting time series of precipitation over the Ob and Lena catchment areas with a running 19 year meanshows for the Ob a steady decline of precipitation since the early 1950s while the Lena keeps its mean nearly for the whole period. This might be the reason for the negative correlation between both rivers shown in Figure 2a and b since 1970, when the area mean precipitation drops below its long term mean.

The main outcome in the paper is that the Ob discharge variability and that of the Lena are not related to each other. The question is of course why do we expect such a relation. If both rivers would have a common forcing that could be expected. A strong large scale forcing all over the world is ENSO, which even reaches the Baltic Sea but

[Figure]

does not seem to reach the 2 rivers. My attached plots FigK3 and FigK5 show clearly that when there is a major event of precipitation in one of the catchment areas, it is likely that the whole catchment area is affected but restricted to one catchment area, with no connection to the other catchment area perhaps enhaced by the mountain range between them. With this statement one has to be careful as one has to take the method of precipitation analysis into account. The method looks for each grid point into 4 directions to find the nearest observational station and makes then a weighted (by the distance) average. In FigK7_stat_dens one can see that the station density, provided by GPCC, in Siberia is very low, especially between both catchment areas, so all grid points within one catchment area will get higher weights to observations within that catchment area. This figure explains as well why in FigK3 and K5 the restriction to the catchment area is stretching for the Ob past the Ural and for Lena towards the SE as these are areas with a higher station density.

Coming back to why do we expect a relation between both rivers? I think it is our inability to imagine the wast extends of the areas in Siberia. Already Napoleon and Hitler fell victim of this inability even for Eurpean Russia.

Here some comments in detail:

page 3 line7: Salehard in the datset by Duemenil et al it is called Salekhard , also looking at google map only Salekhard is known

page 4 line 6: you could mention that in

Arpe K., Leroy S. A. G., Wetterhall F., Khan V., Hagemann S. and H. Lahijani: Prediction of the Caspian Sea Level using ECMWF seasonal forecasts and reanalysis. Theor Appl Climatol DOI 10.1007/s00704-013-0937-6, 2013 the Volgariver discharge has been successfully estimated from the water budget calculations using ERAinterim data and there they use a minimum of 3 month delay between precipitation events and riverdischarge events, longer in winter.

page 7 line 11: do you really mean dumping not damping?

---

## Author Comment (AC1) · 19 Dec 2017

Dear Dr. Klaus Arpe,

The paper tries to find connections between the river discharges of 2 main rivers in Siberia and the atmospheric circulation. It is well written and I enjoyed reading it and found is inspiring. I would recomment accepting it for printing it even if it far from reflecting a breakthrough in science.

Thank you very much for your review comments on the original manuscript. We have revised the manuscript according to your comments. Our point-by-point replies are as follows.

I would add a plot in the introduction of topograpy and catchment areas to set the scene, like the attached file Fig_K1_oro. The Fig.1 provided in the manuscript, showing nearly the same, has too much information and the essentials are not easily to be seen.

As advised, we deleted vector and remade Figure 1 simply.

I further suggest to add some Teleconnection maps (correlation maps) between the discharge or the precipitation over the catchment areas and the precipitation at each grid point as attached File figK2.jpg and FigK3.jpg for the Lena and figK4.jpg and figK5.jpg for the Ob. as suggested in the manuscript, I averaged the precipitation from June to September for such a comparison with the August to November river discharge. It shows nicely the catchment areas (figK2 and FigK4), though the higher correlations between river discharge and precipitation extend a little further, for the Lena to the SE and for the Ob beyond the Ural mountains but no connections between the 2 rivers. I am using the anomaly correlation, i.e. taking away a long term mean from each time series because in meteorology we are mostly interested in the deviations from the climatological mean. For convenience I use the mean of the whole time series

provided. The values are given in the plots in % for clarity with less digits to be printed. I wonder if the definition of correlation has changed during the last 40 years, since I wrote most of my programs. Modern papers show always very high values of > 0.9 while here the best values are> 0.7 and in most of my publications I am happy to reach

5    correlations of up to 0.4.

As you confirmed by the horizontal maps, the variation in P over the Lena river basin is basically related to the R of the Lena and the Ob river is the same when using data during the whole period. That is the same as the positive correlation between the P and R as shown in previous studies (Fukutomi et al. 2003, Serreze et al. 2003, Oshima et al. 2015). However, the

10   first point here is a difference in the relationship (correlation) between the Lena and Ob in each of the epochs. That means that the Rs/Ps of Lena and Ob sometimes show negative correlation and occasionally positive correlation or no correlation in the other period (Figure 2). Fukutomi et al. (2003) revealed the strong negative correlation between the Ps/Rs of the Lena and Ob Rivers during 1980s to mid-1990s. The associated precipitation anomaly maps were shown in

15   Figure 6 of Fukutomi et al. (2003), and we described about those in the third paragraph of the Introduction. The second point of our finding in this study is that such negative correlation frequently seen during the past two centuries based on the tree-ring-reconstructed Rs (Figure 2c) and then we discussed about the associated atmospheric circulation over the region. About those points, we modified some descriptions in the Abstract and the first paragraph of the Summary.

20

I tried also to reproduce the anti-correlation between the Ob and Lena in the 1970/1980s, but could not find anything in the precipitation like that although Fig.2b of the manuscript shows quite a few events like that. Plotting time series of precipitation over the Ob and Lena catchment areas with a running 19 year mean shows for the Ob

25   a steady decline of precipitation since the early 1950s while the Lena keeps its mean nearly for the whole period. This might be the reason for the negative correlation

between both rivers shown in Figure 2a and b since 1970, when the area mean precipitation drops below its long term mean.

The time period of the anti-correlation (negative correlation) is not in the 1970/1980s, but from 1980s to mid-1990s. When we remove the 19-year running mean from the time-series of Figure 2a-c, the correlations (black lines in Figure 2) do not change so much, as confirmed by the following figures. Additionally, we made minor revisions in Figure 2.

[Figure]

The main outcome in the paper is that the Ob discharge variability, and that of the Lena are not related to each other. The question is of course why do we expect such a relation. If both rivers would have a common forcing that could be expected. A strong large scale forcing all over the world is ENSO, which even reaches the Baltic Sea but does not seem to reach the 2 rivers. My attached plots FigK3 and FigK5 show clearly that when there is a major event of precipitation in one of the catchment areas, it is likely that the whole catchment area is affected but restricted to one catchment area, with no connection to the other catchment area perhaps enhaced by the mountain range between them. With this statement one has to be careful as one has to take the method of precipitation analysis into account. The method looks for each grid point into 4 directions to find the nearest observational station and makes then a weighted (by the distance) average. In FigK7_stat_dens one can see that the station density, provided by GPCC, in Siberia is very low, especially between both catchment areas, so all grid points within one catchment area will get higher weights to observations within that catchment area. This figure explains as well why in FigK3 and K5 the restriction to the catchment area is stretching for the Ob past the Ural and for Lena towards the SE as these are areas with a higher station density.

Related to the previous second comment, the main outcome of this study is not the no relationship between the Lena and Ob, but the Rs of the Lena and Ob frequently show negative correlation. While, when analyzing during the whole period, the R of the Lena (Ob) related to the P over the Lena (Ob) river basin, but the Ps over the Lena and Ob sometimes indicate a negative correlation as in the cases of 1980s and mid-1990s. Please see Figure 6 of Fukutomi et al. (2003). About the main outcome of our study, we modified some descriptions in the Abstract and the first paragraph of the Summary.

As you mentioned, in terms of the observation station density over Siberia, it is a concern and discussed in previous studies (e.g., Serreze and Barry, 2005: The Arctic Climate System.

Cambridge University Press, Section 6.1, 148-152). Oshima et al. (2015) investigated the correspondence among components of the water balance by using several datasets based on the independent data sources. We compared discharges from the observation station nearest the river mouth, P-E estimated from meteorological data (specific humidity and wind) of several

5    reanalyses on the basis of the atmospheric water balance method, and P based on satellite and station observations for the three Siberian rivers (Lena, Yenisei, Ob). The results indicated good correspondences in balance and variation. The long-term averages of R and P-E were comparable in magnitude and the P was strongly correlated with R and P-E for each of the individual rivers. Of course, we should be a careful to discuss quantitatively about the P, but the

10    above results indicated that the P dataset from the GPCC and other precipitation products (e.g., PREC/L, APHRODITE) are useful to examine the interannual variation in this region. We added some explanation about the P dataset in the second paragraph of Section 2 "Data and analysis methods".

15    Coming back to why do we expect a relation between both rivers? I think it is our inability to imagine the wast extends of the areas in Siberia. Already Napoleon and Hitler fell victim of this inability even for Eurpean Russia.

It is not so difficult to understand that. We demonstrated that, over the summertime Siberia, the east-west seesaw pattern of large-scale atmospheric circulation frequently emerges as natural

20    internal variability. This east-west seesaw pattern affects opposite influence on the Ps over the Lena and Ob. When the cyclonic anomaly emerges over eastern Siberia, that atmospheric circulation anomaly induces convergence of moisture flux and then increases the P and R of the Lena river. Simultaneously, the anticyclonic anomaly emerges over western Siberia and induces divergence of moisture flux, then decreases P and R of the Ob river, and vice versa. This results

25    in out-of-phase of the Ps/Rs between the two rivers. We modified the explanation in the third paragraph of the Introduction.

Here some comments in detail:

page 3 line7: Salehard in the datset by Duemenil et al it is called Salekhard, also looking at google map only Salekhard is known

5      As described in the manuscript, we used the ArcticRIMS data where the station is named as "Salehard" and we employed that name instead of "Salekhard". Because it is easy to find the used data on the ArcticRIMS website (http://rims.unh.edu/data/station/list.cgi?col=1).

page 4 line 6: you could mention that in

10     Arpe K., Leroy S. A. G., Wetterhall F., Khan V., Hagemann S. and H. Lahijani: Prediction of the Caspian Sea Level using ECMWF seasonal forecasts and reanalysis. Theor Appl Climatol DOI 10.1007/s00704-013-0937-6, 2013 the Volga river discharge has been successfully estimated from the water budget calculations using ERA interim data and there they use a minimum of 3 month delay between precipitation events and

15     river discharge events, longer in winter.

       Thank you for introducing other example of seasonal time lag between R and P. This kind of seasonal time lag may be seen in seasonally frozen rivers. We referred this paper in the last paragraph of Section 2 "Data and analysis methods".

20     page 7 line 11: do you really mean dumping not damping?

       I made a typo. That is "damping". Thank you very much for your careful review.

---

## Short Comment (SC1) · 5 Feb 2018

**Manuscript:** Influence of atmospheric internal variability on the long-term Siberian water cycle during the past two centuries

**Major remarks**

The authors analyse the different long-term behaviour of precipitation and river runoff over the Lena and Ob catchments. Their analysis uses observations, GCM simulations and reconstructed discharges based on tree rings. They could link the anti-correlated behaviour during some periods to an east-west seesaw pattern that seems to be a feature of the general large-scale circulation and the atmospheric internal variability. The study is interesting and provides robust results due its combination of various observation and model data sources.

I only miss some more embedding of the results into the present day climate research. What is the reason for the seesaw pattern? Is there a larger scale process that creates this pattern? Is the seesaw pattern, e.g., related to the circumglobal wave train found by Ding and Wang (2005) in the northern hemispheric during boreal summer? They pointed out that this pattern can favour co-varying patterns of rainfall anomalies over South and East Asia.

Ding, Q., and B. Wang (2005), Circumglobal teleconnection in northern hemisphere summer, J. Climate, 18, 3482-3505.

As the seesaw pattern and the anti-correlation is a real climate feature, do you it can be used as an index to evaluate the performance of GCMs or ESMs? If yes, you may suggest how in the conclusions section?

In section 3, skewnesses are shown in Fig. 3b and Table 2, but it is motivated neither why they are shown nor what the skewness results mean in the context of the present study. If there is not a clear benefit for the study, they may be removed.

I suggest accepting the paper for publication after some revisions have been conducted.

I don't wish do stay anonymous, Stefan Hagemann

**Minor remarks**

In the following suggestions for editorial corrections are marked in *Italic*.

p.1 – line 9
… Ocean, *whereat* the …

p.1 – line 16
… (AGCM) and *fully coupled atmosphere-ocean GCMs* conducted …

p.2 – line 11
*Regarding the* interannual …

p.2 – line 12
… due to *the* large …

p.3 – line 5

… 3 (CMIP3*; Meehl et al. 2007*).

*Meehl, G. A., Covey, C., Delworth, T., Latif, M., McAvaney, B., Mitchell, J. F. B., Stouffer, R. J., and Taylor, K. E.: The WCRP CMIP3 multi-model dataset: A new era in climate change research, Bull. Amer. Meteor. Soc. 88, 1383-1394, 2007.*

p.3 – line 10
It is written:
"Because of limitations on the time period …"

This statement is probably not, what you really mean. In my opinion, the period 1936-2009 of the discharge observation is already quite long. It is probably more that you would like to have even more data to reduce the noise to find significant patterns of variability. Then, you should write this more clearly.

p.3 – line 24/25
…control simulation *is* the … …resolution *is* about … …and the vertical *discretization comprises* 20 layers ...

p.4 – line 4
… R *comprises annual values*, we …

p.4 – line 5
… P *has* large …

p.4 – line 7
*Using a* similar method *as* Tachibana ...

p.4 – line 9
…*2009 are* ..

p.5 – line 31
… (EOF1) *is* the ..

p.7 – line 9
It is written:
"The results in simulations give us several more implications for …"
Strange sentence/English. Please rewrite

p.7 – line 11
What do mean with "dumping"? Please rewrite more clearly.

p.7 – line 24/25
… warming *(Solomon et al. 2007; IPCC 2013).*

Solomon, S., D. Qin, M. Manning, M. Marquis, K. Averyt, M. M. B. Tignor, H. L. Miller Jr., and Z. Chen, Eds. (2007), Climate change 2007: The physical science basis, Cambridge University Press, 996 pp.

IPCC (2013), Climate Change 2013: The Physical Science Basis. Contribution of Working Group I to the Fifth Assessment Report of the Intergovernmental Panel on Climate Change [Stocker, T.F., D. Qin, G.-K. Plattner, M. Tignor, S.K. Allen, J. Boschung, A. Nauels, Y. Xia, V. Bex and P.M. Midgley (eds.)]. Cambridge University Press, Cambridge, United Kingdom and New York, NY, USA, 1535 pp, doi:10.1017/CBO9781107415324.

Figure 1
I cannot really see the thick gray lines. Please improve figure. Actually, the figure looks quite busy. I suggest making two panels out of it.

Figure 2
I suggest adding lines to show the 95% level of significance.

Figure 4
Green dashed inset boxes are hard to see. Please improve figure.

---

## Short Comment (SC2) · 5 Feb 2018

This manuscript presents a statistical analysis on the relationship of river discharges and precipitations between the Lena and Ob river basins using the reconstructed data sets, AGCM simulation, and CMIP3 fully coupled climate model outputs. The results show a time varying correlations in all three data sets, consistent with previous results using shorter observational data set. The variability of sea-saw pattern between the west and east Eurasian continent is responsible for the decadal variation of the correlation coefficients. The research result is important for understanding Eurasian Arctic water cycle and its decadal variability and long-term changes. The manuscript could

be publishable after a revision as described below:

1. The authors attribute the sea-saw pattern is internal variability, but state it is important for long-term changes. Variability and long-term change are two different concepts, with latter generally describing externally forced trend. I would suggest the authors to separate them in the manuscript. 2. Throughout the manuscript, the authors simply mention negative or positive correlations of R and P. This causes confusion of correlation between R and P or correlation of R or P between Lena and Ob. I suggest the authors to provide complete description on this. 3. The authors analyzed the AGCM and CMIP3 climate model outputs to examine the correlation relationship of R and P between the Lena and Ob rivers. To help readers to better understand the modeling results, I suggest the authors to provide full description which AGCM was used and how surface boundary conditioned were defined, as well as how long time the model simulation was carried out. I also suggest the authors to provide information which CMIP3 models were used in 20C3M and PICTL. 4. When comparing the AGCM and CMIP3 climate model results, the authors state that air-sea interaction acts as a damping factor of sea-saw pattern. It is hard for me to understand this. From my understanding, when the modeled P is closer to the reconstructed R, there should be better correlations between P and R. I suggest the authors to clarify this. 5. In line 6, the authors mention "these variables". It is not clear which variables are. In fact, P has been already included in P-E. 6. In line 13, "terrestrial processes" should be specified. 7. In line 11, it would be better to discuss why analyzing the 5 subsets of the data. 8. In line 16, it needs to be clarified what time period was used to do correlation analysis between GPCC P and R. 9. In line 25, the AGCM resolution of about 300 km seems very low to describe water cycle in the river basins. I suggest authors to provide evidence that such a low resolution still can correctly capture P in the river basins. 10. In line 30, what specific discrepancy occurs between P and R?

---

## Editor Comment (EC1) · M. Crucifix (Editor) · 8 Feb 2018

Dear authors, Dear reviewers,

I would like to thank you all for your contribution to Earth System Dynamics. The authors are kindly requested to post public comments to the review comments which have been addressed (but there is no need to produced a revised manuscript at this stage), after which I will close the discussion, and proceed with the next step. Most likely the article will be sent for review again, but we will take all necessary measures to avoid any further delays in the processing of the article.

---

## Author Comment (AC2) · 26 Feb 2018

Dear Dr. Stefan Hagemann,

**Major remarks**

The authors analyse the different long-term behaviour of precipitation and river runoff over the Lena and Ob catchments. Their analysis uses observations, GCM simulations and reconstructed discharges based on tree rings. They could link the anti-correlated behaviour during some periods to an east-west seesaw pattern that seems to be a feature of the general large-scale circulation and the atmospheric internal variability. The study is interesting and provides robust results due its combination of various observation and model data sources.

Thank you very much for the review comments on the original manuscript. We have revised the manuscript according to your comments. Our point-by-point replies are as follows.

I only miss some more embedding of the results into the present day climate research. What is the reason for the seesaw pattern? Is there a larger scale process that creates this pattern? Is the seesaw pattern, e.g., related to the circumglobal wave train found by Ding and Wang (2005) in the northern hemispheric during boreal summer? They pointed out that this pattern can favour co-varying patterns of rainfall anomalies over South and East Asia.
Ding, Q., and B. Wang (2005), Circumglobal teleconnection in northern hemisphere summer, J. Climate, 18, 3482-3505.

As mentioned in the manuscript, the reason for the east-west seesaw pattern is a summertime atmospheric internal variability over Siberia. The AGCM control simulation demonstrated the seesaw pattern over Siberia in summer. Thus, without external forcing like changes in SST, sea ice, greenhouse gases and solar activity, the seesaw pattern often emerges by chance.

In addition, we discussed about other reasons for the large-scale atmospheric circulation associated with P variability over Siberia. In the last part of Section 4, we described that the remote influence of Atlantic multidecadal variation, quasi-stationary Rossby waves over Eurasia and Arctic dipole anomaly affect the Siberian P based on the previous studies. As you pointed out, the circumglobal wave train may be another candidate. However, those specific effects are not clear in our analysis and future work is needed. We described about these in the last two paragraphs of Section 4.

As the seesaw pattern and the anti-correlation is a real climate feature, do you it can be used as an index to evaluate the performance of GCMs or ESMs? If yes, you may suggest how in the conclusions section?

Yes. In this study, we evaluated the seesaw pattern and negative correlation in the AGCM and CMIP3 simulations on the basis of three ways. As for the negative correlation, we calculated the two statistics of median and skewness for the 15-year running correlations (Sub-subsection 3.1.2). As for the seesaw pattern, we performed an EOF analysis to identify the dominant pattern of large-scale circulation, and then calculated the pattern correlation of EOF2 between the JRA-55 and each of the simulations (Subsection 3.2). As for the relationship between the negative correlation and the seesaw pattern, we defined two indices of $\Delta Z500_{WE}$ and $\Delta P_{LO}$, and calculated the correlation between them (Subsection 3.2).

Our explanations were insufficient and we described about these in the corresponding sections.

In section 3, skewnesses are shown in Fig. 3b and Table 2, but it is motivated neither why they are shown nor what the skewness results mean in the context of the present study. If there is not a clear benefit for the study, they may be removed.

As you know, the skewness is a measure of asymmetry of probability or frequency distribution. Here, we examined the frequency distribution of correlations of P between the Lena and Ob. So, when the correlation is distributed in the negative side, the skewness has positive value. We added this explanation in the second paragraph of Sub-subsection 3.1.1.

I suggest accepting the paper for publication after some revisions have been conducted.

I don't wish do stay anonymous, Stefan Hagemann

Thank you, Stefan.

**Minor remarks**

In the following suggestions for editorial corrections are marked in *Italic*.

Thank you for the careful review.

p.1 – line 9

… Ocean, *whereat* the …

We corrected as suggested.

p.1 – line 16

… (AGCM) and *fully coupled atmosphere-ocean GCMs* conducted …

We corrected as suggested.

p.2 – line 11

*Regarding the* interannual …

We corrected it.

p.2 – line 12

… due to *the* large …

We corrected it.

p.3 – line 5

… 3 (CMIP3*; Meehl et al. 2007*).

Meehl, G. A., Covey, C., Delworth, T., Latif, M., McAvaney, B., Mitchell, J. F. B., Stouffer, R. J., and Taylor, K. E.: The WCRP CMIP3 multi-model dataset: A new era in climate change research, Bull. Amer. Meteor. Soc. 88, 1383-1394, 2007.

We added this reference.

p.3 – line 10

It is written:

"Because of limitations on the time period …"

This statement is probably not, what you really mean. In my opinion, the period 1936-2009 of the discharge observation is already quite long. It is probably more that you would like to have even more data to reduce the noise to find significant patterns of variability. Then, you should write this more clearly.

The 74-year record (1936-2009) of observed R is not long enough for this study. As in Figure 2a, we could find the negative correlation period of the Lena and Ob Rs during 1980s to mid-1990s and the positive correlation period during 1960s to 1970s, one by one. But we couldn't judge whether there is a certain tendency of the correlation based on the 74-year record. On the other hand, we could reveal the tendency of frequent negative correlation based on the 191-year record of reconstructed R. The 111-year record of observed P also show the similar tendency of negative correlation. The time period of negative correlation seems one or two decades. To detect such a tendency of the correlation on decadal timescale, the usage of long-term record is desirable. In addition, to detect a robust tendency of the correlation, we made subset of 150-year records and increased sample size of data. We added some explanation in the first paragraph of Section 2.

p.3 – line 24/25

…control simulation *is* the … …resolution *is* about … …and the vertical *discretization comprises* 20 layers ...

We corrected them.

p.4 – line 4

… R *comprises annual values*, we …

We corrected as suggested.

p.4 – line 5

… P *has* large …

This is not corresponding. The sentence was changed.

p.4 – line 7

*Using a* similar method *as* Tachibana ...

We corrected as suggested.

p.4 – line 9

…2009 *are* ..

We corrected it.

p.5 – line 31

… (EOF1) *is* the ..

We corrected it.

p.7 – line 9 It is written:

"The results in simulations give us several more implications for …"

Strange sentence/English. Please rewrite

We revised this sentence.

p.7 – line 11

What do mean with "dumping"? Please rewrite more clearly.

We made a mistake and "damping" is correct. We added some explanation in the second paragraph of Section 4.

p.7 – line 24/25

… warming *(Solomon et al. 2007; IPCC 2013).*

Solomon, S., D. Qin, M. Manning, M. Marquis, K. Averyt, M. M. B. Tignor, H. L. Miller Jr., and Z. Chen, Eds. (2007), Climate change 2007: The physical science basis, Cambridge University Press, 996 pp.

IPCC (2013), Climate Change 2013: The Physical Science Basis. Contribution of Working Group I to the Fifth Assessment Report of the Intergovernmental Panel on Climate Change [Stocker, T.F., D. Qin, G.-K. Plattner, M. Tignor, S.K. Allen, J. Boschung, A. Nauels, Y. Xia,

V. Bex and P.M. Midgley (eds.)]. Cambridge University Press, Cambridge, United Kingdom and New York, NY, USA, 1535 pp, doi:10.1017/CBO9781107415324.

We corrected these references.

Figure 1

I cannot really see the thick gray lines. Please improve figure. Actually, the figure looks quite busy. I suggest making two panels out of it.

As the first reviewer of Dr. Arpe pointed out, we deleted vector and remade it simple.

Figure 2

I suggest adding lines to show the 95% level of significance.

We added the lines of the 95% significant level.

Figure 4

Green dashed inset boxes are hard to see. Please improve figure.

We changed the color of inset boxes.

---

## Author Comment (AC3) · 27 Feb 2018

Dear Xiangdong Zhang,

This manuscript presents a statistical analysis on the relationship of river discharges and precipitations between the Lena and Ob river basins using the reconstructed data sets, AGCM simulation, and CMIP3 fully coupled climate model outputs. The results show a time varying correlations in all three data sets, consistent with previous results using shorter observational data set. The variability of sea-saw pattern between the west and east Eurasian continent is responsible for the decadal variation of the correlation coefficients. The research result is important for understanding Eurasian Arctic water cycle and its decadal variability and long-term changes. The manuscript could be publishable after a revision as described below:

Thank you very much for your review comments on the original manuscript. We have revised the manuscript according to your comments. Our point-by-point replies are as follows.

1. The authors attribute the sea-saw pattern is internal variability, but state it is important for long-term changes. Variability and long-term change are two different concepts, with latter generally describing externally forced trend. I would suggest the authors to separate them in the manuscript.

As you pointed out, the long-term change is also important for P and R variabilities. While Fukutomi et al. (2003) and MacDonald et al. (2007) discussed about the long-term variations on decadal timescale, it seems that the long-term changes do not affect the time series of 15-year running correlation in Figure 2. In fact, as replied to Dr. Klaus Arpe's comment, when we remove the 19-year running mean from the raw time-series of P and R in Figure 2a-c, the correlations do not change so much. We added this result in the last part of Section 3.1.1.

2. Throughout the manuscript, the authors simply mention negative or positive correlations of R and P. This causes confusion of correlation between R and P or correlation of R or P between Lena and Ob. I suggest the authors to provide complete description on this.

I agree with you. That point was confusing and we revised the expression clearly throughout the manuscript.

3. The authors analyzed the AGCM and CMIP3 climate model outputs to examine the correlation relationship of R and P between the Lena and Ob rivers. To help readers

to better understand the modeling results, I suggest the authors to provide full description which AGCM was used and how surface boundary conditioned were defined, as well as how long time the model simulation was carried out. I also suggest the authors to provide information which CMIP3 models were used in 20C3M and PICTL.

While the description of the AGCM control simulation was shown in the third paragraph of Section 2, we added further explanation about the 20C3M and PICTL simulations in the fourth paragraph of Section 2.

4. When comparing the AGCM and CMIP3 climate model results, the authors state that air-sea interaction acts as a damping factor of sea-saw pattern. It is hard for me to understand this. From my understanding, when the modeled P is closer to the reconstructed R, there should be better correlations between P and R. I suggest the authors to clarify this.

We examined the relationship between R and P based on the observation and reconstruction. On the other hand, in the AGCM and CMIP3 simulations, we examined the relationship between the P and atmospheric circulation and did not analyze simulated R. While the AGCM control simulation is forced by the fixed boundary conditions, the CMIP3 simulations are based on ocean-atmosphere coupled model and have the effect of air-sea interaction. If possible, it is better to simulate the P and large-scale circulation over Siberia with the same kind of AGCM and coupled GCM. But, unfortunately, we don't have a coupled model and cannot do that. Our discussion in this study is only based on the CCSR/NIES AGCM and CMIP3 simulations. We added further discussion in the second paragraph of Section 4.

5. In line 6 (P. 2), the authors mention "these variables". It is not clear which variables are. In fact, P has been already included in P-E.

We specified the variables (i.e., R and P-E).

6. In line 13 (P. 2), "terrestrial processes" should be specified.

The discharge control via dams, permafrost condition associated with runoff process, distributions of lake, wetland and vegetation associated with evapotranspiration are included in the terrestrial processes. We added these in the text.

7. In line 11 (P. 3), it would be better to discuss why analyzing the 5 subsets of the data.

As in Figure 2, the negative correlations were frequently seen during the past two centuries (Figure 2c) and the time period of the negative correlation seems one or two decades. To detect a robust tendency of the correlation, we made subset of 150-year records and increased sample size of data. We added that explanation in the first paragraph of Section 2.

8. In line 16 (P. 3), it needs to be clarified what time period was used to do correlation analysis between GPCC P and R.

The time period of the correlation is from 1901 to 2010. We described it.

9. In line 25 (P. 3), the AGCM resolution of about 300 km seems very low to describe water cycle in the river basins. I suggest authors to provide evidence that such a low resolution still can correctly capture P in the river basins.

As you pointed out, the resolution of our simulation is lower than in the recent AGCM/GCM's studies. In the previous studies, however, Numaguti 1999 and Kurita et al. 2005 examined precipitation recycling and source of precipitating water over Eurasia using an AGCM with T42 spatial resolution same as in our simulation. They indicated that the spatial pattern and seasonal cycle of simulated P and P-E over Eurasia are generally consistent with the observed features in the seasonal timescale. In this study, observed features of the negative correlation of P between eastern and western Siberia, the east-west seesaw and the relationship between the negative correlation and seesaw pattern were reproduced in the AGCM simulation. Therefore, this resolution of about 300km is enough for the purpose of this study. We added this explanation in the third paragraph of Section 2.

Numaguti, A. (1999), Origin and recycling processes of precipitating water over the Eurasian continent: Experiments using an atmospheric general circulation model, J. Geophys. Res., 104(D2), 1957–1972, doi:10.1029/1998JD200026.

Kurita, N., A. Sugimoto, Y. Fujii, T. Fukazawa, V. N. Makarov, O. Watanabe, K. Ichiyanagi, A. Numaguti, and N. Yoshida (2005), Isotopic composition and origin of snow over Siberia, J. Geophys. Res., 110, D13102, doi:10.1029/2004JD005053.

10. In line 30 (P. 4), what specific discrepancy occurs between P and R?

There are some error and uncertainty for both the observed P and reconstructed R and they result in the discrepancy between the P an R. The observation stations of P are sparse in Siberia and there is difficulty in the P measurement such as wind-induced undercatch, wetting, and evaporation losses. These make an error and uncertainty for the P. The long-term R during the

past two centuries is reconstructed based on the tree-ring width. While the tree-ring width has an indirect relation with the R, the both are mainly related through the P. There are also other influences such as SAT, solar radiation, nitrogen and so on. In addition, the tree-ring width is affected by meteorological conditions during the growing season in summer and there must be less contribution from the P during winter. As a result, the reconstructed Rs could explain 43% of the observed variability for the Lena and 51% for the Ob (MacDonald et al. 2007). We added some explanation in the second paragraph of Sub-subsection 3.1.1.

---

## Author Response (AR2)

Dear Dr. Michel Crucifix,

Thank you very much for your careful check for our revised manuscript.
We revised the manuscript in line with your comments.

Best regards,
Kazuhiro Oshima